# Association of Cardiovascular Disease Risk and Health-Related Behaviors in Stroke Patients

**DOI:** 10.3390/ijerph20043693

**Published:** 2023-02-19

**Authors:** Rezarta Lalo, Ilirjana Zekja, Fatjona Kamberi

**Affiliations:** 1Department of Health Care, Faculty of Health, University of Vlora “Ismail Qemali”, L. Pavarësia, 9400 Vlorë, Albania; 2Faculty of Technical Medical Sciences, University of Medicine Tirana, 8RRM+W7X, Rruga e Dibrës, 1001 Tirana, Albania; 3Research Centre for Public Health, Faculty of Health, University of Vlora “Ismail Qemali”, L. Pavarësia, 9400 Vlorë, Albania

**Keywords:** association, better health, cardiovascular disease risk, health-related behaviors, stroke patients

## Abstract

Brain stroke continues to be a leading cause of mortality and disability in both developed and developing countries, with higher healthcare costs due to the long-term care and rehabilitation that it incurs. The purpose of the current study was to assess the association between brain stroke patients’ health-related behaviors and their risk for cardiovascular disease. Methods: A cross-sectional study was carried out from March to August 2022 in the Vlora district regional hospital in Albania. The study included 150 out of 170 participants who met the necessary criteria, achieving an 88% response rate. Measurement tools included the Framingham Cardiovascular Risk Scale (FRS) and the Lifestyle Health Promotion Profile II (HPLP II). Results: The patients’ average age was 65.9 ± 9.04 years. Over 65% of the stroke patients suffer from diabetes, and 47% from hypertension. About 31% of them have a high risk of hyperlipidemia (mean TC = 179 ± 28.5). About 32% of the brain stroke patients manifested unhealthy behaviors, while 84% of them had a high risk of cardiovascular disease (FRS = 19.5 ± 0.53). Cardiovascular disease (CVD) risk was statistically associated with stress management behaviors (*p* = 0.008; OR = 0.20; CI = 95%). This risk was highest in the over-70 age group as well as in men. Conclusion: Brain stroke patients had a high probability of developing CVD. For better health among stroke patients, new evidence-based behavior change approaches must be introduced into preventative and management programs.

## 1. Introduction

An attack on the brain or stroke is a neurological vascular event that severely damages the central nervous system. Approximately 613,148 strokes were reported in all EU nations in 2015, and it is projected that the number will increase by 34% in 2035 as a result of population aging and increased risk factors [1]. Brain attacks are the second biggest cause of mortality and disability, according to the OECD (2018), making them a significant public health concern [2].

Patients with brain stroke appear to have a significant prevalence of cardiovascular disease (CVD) [3]. Previous studies have identified a number of both non-modifiable and modifiable risk factors, including sociodemographic characteristics, medical history, prior stroke history, hypertension, hypercholesterolemia, diabetes mellitus, psychosocial stress, and unhealthy lifestyle behaviors [4]. According to worldwide standards and research, it is strongly recommended to promote activities that specifically target modifiable factors, with an emphasis on brain stroke prevention among the population [5].

Multimodal behavioral therapies, such as control of clinical parameters, modification of adverse lifestyles, and adherence to preventive medications have been shown to improve the health of stroke survivors, according to the results of a randomized controlled trial [6].

Furthermore, though the burden of brain stroke is increasing in low and middle-income countries, attempts to avoid it within the medical system are still in their early stages [7].

Albania, an emerging country in the South-East Europe (SEE) region, inspires to be part of the European Union and is characterized by an epidemiological transition with significant changes in the lifestyle and behavioral characteristics of the population. The latest World Health Organization (WHO) statistics from 2017 show that 4675 brain stroke deaths occurred, accounting for 24.15% of all deaths in Albania. The age-regulated death rate in this developing country is 105.31 per 1000 persons, which is a major cause for concern for health-care experts [8].

However, there is little information available on Albania’s specific cerebrovascular disease burden. There are particularly few scientific studies that describe the prevalence and incidence of these disorders in the adult Albanian population [9]. One study, conducted in the Albanian city of Vlora as a diabetes and hypertension screening programme that specifically focused on the risk assessment of stroke in a sample of randomly selected adult persons, discovered that the risk for stroke was average for 38% of participants and relatively high for 24% of them. While stroke risk, levels of arterial blood pressure, and diabetes were statistically significant [10]. Meanwhile, studies in relation to strategies or different management risks related to brain stroke-related behaviors are lacking. Few findings in this regard imply that quick risk assessment for diabetes and brain stroke is an affordable early screening technique that can be implemented in primary healthcare, especially in countries with limited tools and resources [11].

In addition, one study suggests that there is a significant knowledge gap about the efficacy of different brain stroke prevention strategies in developing countries. For example, it is unclear which interventions would be appropriate and cost-effective in reducing the number of strokes in these countries. Poor knowledge, a lack of information, and a lack of preventative health-related behavior remain some of the greatest challenges in public healthcare, preventing stroke patients from accessing state of the art treatment [7].

The important findings above highlight the need to investigate risk factors and health-related behaviors. Furthermore, the implementation of behavior modification strategies for brain stroke prevention in Albania may be aided by understanding the factors that affect health-related behaviors. The purpose of the current study was to accurately assess the association between brain stroke patients’ health-related behaviors, such as smoking cigarettes, drinking alcohol excessively, unhealthy diet, lack of exercise, and stress, and their risk for cardiovascular disease.

## 2. Materials and Methods

### 2.1. Study Area, Study Design, and Study Period

A cross-sectional study was conducted in the regional hospital pathology department in Vlora district, Albania, between March and August 2022.

### 2.2. Sample Size Estimation and Sampling Technique

A non-probability consecutive sampling technique based on inclusion and exclusion criteria was selected for data collection. The *inclusion criteria* included all hospitalized patients in the pathology ward that were diagnosed with brain stroke by magnetic resonance imaging (MRI) or scanner (computed tomography CT scan), were over the age of 18 years, with demonstrated signs of neurological stability (the result of an examination of mental state was more than 21) and that were eligibility to participate in the study. The *exclusion criteria* were patients with subarachnoid hemorrhage, severe aphasia, dementia, or severe comorbidities (e.g., end-stage renal disease, Parkinson’s disease or multiple sclerosis). The Cochran formula: n = (Za)^2^ (p × q)/d^2^ was used to determine the sample size, where p = brain stroke prevalence (according to Albanian studies, it is about 15%), q = prevalence complement, error margin = d, alpha = 85 percent significance level, and n = 2.85 × 2.85 (0.15 × 0.85)/0.1 × 0.1 = 138. This formula determined that 138 people were the ideal sample size. Finally, 150 participants who met the criteria and expressed a willingness to participate were included out of 170 patients who were hospitalized throughout the duration of the study, resulting in an 88% response rate. All participants provided informed consent.

### 2.3. Data Collection and Study Tools

Information from medical records was used to assess medical history, drug treatment, and lipid and glucose profile data. Two health professionals measured blood pressure and body mass index (BMI) during their daily clinical practice and had previously agreed to cooperate in data collection. To assess risky behaviors including alcohol consumption and smoking habits, we followed the respective WHO Framework Convention on Tobacco Control Guidelines [12] and the Canada Low-Risk Alcohol Drinking Guidelines [13]. High-risk behaviors are characterized as actions that raise the possibility of contracting a disease or suffering an injury, which can then result in disability, death, or societal issues. Violence, alcoholism, tobacco use disorder, risky sexual conduct, and eating disorders are among the most prevalent high-risk behaviors [14]. We used the Health Promotion Lifestyle Profile (HPLP II), a self-reported scale [15] that evaluates health-related behaviors in six areas. We selected three of the six HPLP II dimensions to assess the physical, nutrition, and stress management activities of participants in the past week before their stroke using a four-point Likert scale. Based on HPLP II scoring, each response ranged from 4 to 32 points for physical activity and stress management and a score of 4 to 36 points for nutrition. The total score ranged from 12 to 100 points. A score for an overall health-promoting lifestyle was obtained by calculating a mean of the individual’s responses to all 25 items for three subscales. Three subscale scores were obtained similarly by calculating the mean of the responses to subscale items. Lifestyle behaviors were classified into two categories: 12–50 points = incorrect behavior and correct behavior = 51–100 points.

To evaluate comorbidities, the Comorbidity Function Index (CFI) was applied [16]. Every comorbidity listed in the index receives the same weighting: 1 point if present, 0 point if absent. The final score, which might be between 0 and 18 points, was the total of all criteria. A higher score was derived by adding the “yes” responses, which then indicated worse comorbidity. In addition, the questionnaire asks users to self-report sociodemographic data. We used the Framingham Risk Score (FRS) to further examine the likelihood of developing cardiovascular disease. The six coronary risk factors, including age, gender, total cholesterol (TC), high-density lipoprotein (HDL) cholesterol, systolic blood pressure, and smoking behaviors, were used to construct FRS scores [17]. The FRS calculates the 10-year probability of clinical stroke manifestation. Three categories of CVD risk percentage were established: low risk (10%), moderate risk (>10%), and high risk (>20%).

### 2.4. Statistical Analysis

The SPSS-23 statistical package was used to analyze the data. We used descriptive analyses for clinical profiles, sociodemographic factors, and health- related behaviors.

The mean, minimum, maximum, and standard deviation were calculated for continuous variables assessed using points (behavior, risk), whereas frequencies and percentages were calculated for categorical variables. The Chi-square probability was used to demonstrate significant changes in the values of the dependent variable based on the levels of an independent variable, and the value *p* < 0.05 was accepted as statistically significant. Analysis of variance (ANOVA) was also used for quantitative variables. Where a significant statistical difference was observed, the analysis was continued by regression analysis. For qualitative variables (physical activity, nutrition, and stress management), univariate logistic regression was used—wherein the compared probabilities (OR) were analyzed—while for quantitative variables (such as BMI, systolic blood pressure, LDL, HDL, TC, triglycerides (TG), fasting glucose (FG), HbA1c, smoking status, and alcohol consumption), simple linear regression was used.

## 3. Results

### 3.1. Socio Demographic Characteristics and Risk Profile of Stroke Patients

One hundred and fifty patients, with a mean age of 65.9 ± 9.04 and composed of a majority of men (64%), completed the questionnaires. The risk profile of the individuals revealed a tendency for obesity, that the majority of the patients had diabetes, that a significant percentage had a greater risk for hyperlipidemia and were also considered hypertensive, and that the patients were both smokers and drinkers (Table 1).

### 3.2. Evaluation of the Health-Promoting Lifestyle of Participants

Referring to the findings in Table 2, about 32% of brain stroke patients manifested unhealthy lifestyle behaviors. Overall, the HPLP II score was statistically significantly associated with only two sociodemographic variables, such as age and gender (Table 3). It was observed that males and patients in the age group 51–60 years old manifested healthier behaviors compared with other categories.

### 3.3. Predictors of Cardiovascular Disease Risk (FRS Score)

The mean FRS score was 19.5 ± 0.53 (Table 1). The results presented in Table 4 found a statistically significant relationship between FRS score and variables such as age, gender, and stress management. It is noted that the majority of patients belonging to the age group of 61–70 years old and 65% of women are more likely to be exposed to the risk of cardiovascular disease.

## 4. Discussion

According to O’Donnell (2010), a risk profile increases the risk of having a brain attack or stroke. To lower the incidence of stroke, it is crucial to analyze the risk profile [4]. The findings of our investigation also support these data. The over 61 age group, men, and people from low socioeconomic backgrounds had the highest incidence of stroke. According to certain studies [18,19,20,21], the incidence increases with age; however, other investigations have revealed that the incidence increases even at younger ages [22].

Regarding the other demographic characteristics mentioned, our results are in line with other studies conducted in Albania and other countries [9,21], which demonstrate that the risk of stroke is higher in men and patients with a lower socioeconomic level. Similarly, it was found that people of a lower socioeconomic level are more likely to have stroke due to their inability to receive quality hospital and rehabilitation services [23,24]. Related to gender factor, studies have shown that this can be explained by the protective role of estrogen in women’s cerebral circulation [25] and the detrimental effects of risk factors including smoking and ischemic heart disease, which are more common in men [26]. Future research should be undertaken to better understand the biochemical causes of stroke in each gender, according to these findings.

Our results, which are in line with those of other studies, demonstrate that diabetes mellitus, hypertension, and hyperlipidemia were significant predictors of the risk profile of patients with a high prevalence of stroke, primarily in those over 60 years of age. This suggests an urgent need for improved treatment for patients with hypertension and diabetes, as well as early blood control and a focus on their fat profiles [19,27,28]. The BMI assessment results of stroke patients are interesting since they reveal that the majority of participants were overweight and that a large fraction was obese. Globally, obesity is beginning to be regarded as a factor that affects the death rate from stroke, and our study’s findings are concerning in terms of risk exposure and are consistent with other studies [21,29]. Additionally, studies have revealed that hypertension is related to ischemic and hemorrhagic stroke, while obesity is related to the cardiovascular risk that results in ischemic stroke [19,20,21]. The above relationship can be explained by the fact that the amount of adipose tissue in obese patients leads to subsequent atherosclerosis and cerebral vessel occlusion resulting in ischemic stroke. High blood pressure can also damage blood vessels inside the brain, causing bleeding in the brain [22,23].

In recent years, numerous research has investigated the effects of consuming alcohol and smoking on the prevalence of stroke [15,18,30]. This is also visible in the study’s findings, which show that 11.6 ± 5.6 clouds of smoke and 3.5 ± 1.6 drinks are consumed daily, respectively.

Brain stroke patients reported statistically significant associations between sociodemographic characteristics and an average level of healthy habits. Patients under 60 years old showed healthier behaviors. Surprisingly, men displayed healthier practices than women. Based on studies, sociodemographic variables have been found to have an impact on different health habits. In contrast with our findings, another study found that patient health behaviors increased with age [31], while another found that older patients with higher educational levels demonstrated better health behaviors [32]. In contrast with our results, a study found that the behaviors of the female subjects were better [28]. This range of data shows that educational initiatives should focus on the most vulnerable populations. Furthermore, our study provides evidence for the subscales of health behaviors selected as most related to self-efficacy and management [15]. The results show that the nutrition subscale (20.6 ± 4.48) and stress management (17.5 ± 5.11) had the highest mean scores, while the physical activity subscale (16.5 ± 5.4) had the lowest. Jiang et al. (2014) discovered that the physical activity subscale had a lower score [33], similar to our study, while hypertensive stroke patients reported healthier habits related to the nutrition subscale, with higher mean scores reported in another study [28]. These findings have significant therapeutic practice implications that can help to further encourage patients to adopt healthier habits.

Recent scientific research has investigated the relationship between unhealthy lifestyle behaviors and the risk of cardiovascular events [34,35]. Referring to a meta-analysis, one study determined that long-term sedentary behavior and physical inactivity increased the risk of cardiovascular diseases (CVD) in healthy adults [34]. In addition, another study provided evidence concerning health promoting behaviors (HPB) and risk of cardiovascular events among patients with CVD. Findings in this study reveal that, among HPB subscales, only nutrition indicated a significant correlation with risk of cardiovascular events [35]. Furthermore, the assessment of cardiovascular risk in stroke patients has been proven to have a positive influence on primary and secondary prevention, according to a systematic review of the literature [3].

To the best of our knowledge, this is the first study to evaluate the risk of cardiovascular disease in brain stroke patients using the Framingham scale in Albania. As stated in the results section, the mean FRS (19.5 ± 0.53) was high, and 84% of stroke patients had a high risk of CVD. Only the subcategory of stress management health behaviors showed a significant association in the regression analysis to identify CVD risk factors. According to our study, the overall Health Promoting Lifestyle Profile II (HPLP-II) did not affect the degree of cardiovascular risk of stroke patients, a finding similar to a study conducted previously [31]. The above study [31] also found that cardiovascular risk was only statistically associated with the subcategory of physical activity, demonstrating the advantages associated with including physical activity in prevention programs. Patients who participated in physical activity had a decreased risk of CVD. The reverse is true, as shown by our findings, which point to the urgent need to incorporate physical activity or a healthy diet into future educational programs and services, as well as its impact on lowering cardiovascular risk in stroke patients [3,32,33].

### Strength and Limitations

This study has several limitations. Since we only evaluated patients who had moderate strokes when admitted to the pathology ward, our findings cannot be applied to those who had mild strokes or those who were discharged from the hospital. In addition, the cross-sectional design of the study makes it difficult for us to identify the underlying causes of the correlation between the variables. Blood pressure was measured during hospital stays, meaning that this measurement may not accurately reflect blood pressure before a stroke, and does not take into account how drug administration may have affected the actual value of blood pressure. However, we have taken into account the list of previous medications when calculating the CVD risk. In addition, some examinations that depend on self-report measures could be biased. Despite these limitations, the study has the advantage of using standardized tools whose validity and reliability have been examined in prior research. Another benefit and strength of the study is the use of clinical data from patient records and other digital examinations (CT scanners). The power of the study, which is the first of its kind in Albania, is strengthened by the completion of anthropometric and arterial pressure measurements by professional nurses. Future research is necessary to evaluate the effects of interventions in which nurses have a primary role in the modification of health and behavioral risk factors.

## 5. Conclusions

Diabetes mellitus, hypertension, and hyperlipidemia were shown to be significant predictors in the risk profiles of brain stroke patients in our study based on clinical and anthropometric data. Being overweight, abusing alcohol, and using tobacco were all considered to reinforce this profile. Patients of a younger age and of the male gender reported healthier behavior. According to the FRS scale, good self-management of stress was associated with a lower cardiovascular risk score. These findings imply that healthcare professionals should include cardiovascular disease risk factors in addition to stroke risk profiles when assessing stroke patients’ training and education needs.

## Figures and Tables

**Table 1 ijerph-20-03693-t001:** Sociodemographic characteristics and risk profile of stroke patients.

Variables	n (%)	Mean ± SD
*Sociodemographic variables*		
Age Age by gender	-	65.92 ± 9.04 68.167 ± 8.73 (female) 64.656 ± 9.01 (male)
Gender (female)	54 (36)	-
Education (less than high school)	103 (69)	-
Living (alone)	10 (7)	-
Economic status (moderate)	86 (57)	-
Residence (village)	52 (35)	-
*Risk profile*		
CFI (score)	-	2.5 ± 1.63
FRS (score)	-	19.5 ± 0.53
BMI (kg/m^2^)	-	27.2 ± 3.16
Systolic BP (mm Hg)	-	164.307 ± 27.06
Diastolic BP (mm Hg)	-	90.389 ± 12.26
BP over 140/90	70 (46.7)	-
BP over 180/110	38 (25.3)	-
LDL (mg/dL)	-	140.1 ± 23.8
LDL > 160 mg/dL	31 (20.7)	
HDL (mg/dL)	-	40.4 ± 15.4
HDL < 35 mg/dL	44 (29.3)	-
TC cholesterol total (mg/dL)	-	208.9 ± 54.3
TC > 240 mg/dL	46 (30.7)	-
TG (mg/dL)	-	165.5 ± 63.5
TG > 200 mg/dL	24 (16)	-
FG (mg/dL)	-	178.1 ± 89.0
FG > 126 mg/dL	107 (71.30)	-
HbA1c %	-	8.5 ± 12.8
HbA1c > 6.5%	97 (64.7)	-
Smoking status over the past 7 days (yes)	70 (46.6)	-
Number of cigarettes per day	-	11.67 ± 5.67
Alcohol consumption over the last 7 days (yes)	47 (31.3)	-
Number of drinks per day	-	3.57 ± 1.67

**SD**: Standard deviation, **FRS**: Framingham Risk Score, **BMI**: body mass index, **BP**: blood pressure, **HDL**: high-density lipoprotein, **LDL**: low-density lipoprotein, **TC**: total cholesterol, **TG**: triglycerides, **FG**: fasting glucose, **HbA1c**: hemoglobin A1c, **CFI**: Comorbidity Function Index.

**Table 2 ijerph-20-03693-t002:** Evaluation of health promoting lifestyle of participants.

Variables	n (%)	Mean ± SD
**Overall HPLP II**		
Correct	102 (68.0)	54.67 ± 13.34
Incorrect	48 (32.0)	
*Total*	150 (100.00)	
**Physical activity subscale**		
Correct	98 (65.3)	16.50 ± 5.40
Incorrect	52 (34.7)	
*Total*	150 (100.0)	
**Nutrition subscale**		
Correct	119 (79.3)	20.62 ± 4.48
Incorrect	31 (20.7)	
*Total*	150 (100.0)	
**Stress management subscale**		
Correct	108 (72.0)	17.50 ± 5.11
Incorrect	42 (28.0)	
*Total*	150 (100.0)	

**HPLP II**: Health Promotion Profile II, **SD**: Standard deviation.

**Table 3 ijerph-20-03693-t003:** Association between health promoting lifestyle and sociodemographic factors.

Predictors	Overall HPLP II						
		*p*	ANOVA		Regression Analysis		
Sociodemographic	HPLP II		F	F Critical	OR	95% CI	
Factors	Mean ± SD					Lower Bound	Upper Bound
*Age* (*years*)							
Until 50	59.3 ± 15.7	**0.016**	3.544	0.167	2.276	0.418	12.378
51–60	60.3 ± 11.8				4.400	1.466	13.205
61–70	53.6 ± 11.4				1.517	0.695	3.314
>70	51.5 ± 14.9				Reference		
*Gender*							
Female	53.6 ± 13.8	**0.020**	0.493	0.478	0.909	0.446	1.852
Male	55.3 ± 13.1				Reference		
*Education*							
Bachelor	76.5 ± 0.71	0.098	2.137	2.300			
High school	54.8 ± 12.3						
Professional degree	49.5 ± 9.4						
Less than high school	54.5 ± 13.7						
*Living*							
With someone	55 ± 13.1	0.298	1.099	2.110			
Alone	50 ± 13.3						
*Economic status*							
Moderate and high	77 ± 0.01	0.221	1.525	1.670			
Low	53.9 ± 14.1						

**HPLP II**: Health Promotion Profile II, **SD**: Standard deviation, **OR**: odds ratio, **CI**: confidence interval, **ANOVA**: analysis of variance. The bold number are to emphasize the *p* value as significant.

**Table 4 ijerph-20-03693-t004:** Predictors of cardiovascular disease risk (Framingham Risk Score) and regression analysis.

Predictors		CVD Risk					*p* Value
		High, n (%)	Moderate, n (%)	Low, n (%)		Total n (%)	
*Age (years)*							
Until 50		4 (2.7)	3 (2.0)	1 (0.7)		8 (5.4)	**0.001**
51–60		24 (16.0)	10 (6.7)	0		35 (22.7)	
61–70		51 (34.0)	5 (3.3)	1 (0.7)		57 (38.0)	
>70		47 (31.3)	2 (1.3)	2 (1.3)		50 (33.9)	
*Gender*							
Female		35 (23.3)	15 (10.0)	4 (2.7)		54 (36.0)	**0.002**
Male		91 (60.7)	5 (3.3)	0		96 (64.0)	
*Education*							
Less than high school		87 (58.0)	13 (8.7)	3 (2.0)		103 (68.7)	0.853
Bachelor		1 (0.7)	1 (0.7)	0		2 (1.3)	
High school		33 (22.0)	5 (3.3)	1 (0.7)		39 (26.0)	
Professional degree		5 (3.3)	1 (0.7)	0		6 (4.0)	
*Residence*							
Village		47 (31.4)	5 (3.3)	0		52 (34.7)	0.189
City		79 (52.6)	15 (10.0)	4 (2.7)		98 (65.3)	
*Living*							
With someone		119 (79.3)	18 (12.0)	3 (2.0)		140 (93.3)	0.703
Alone		7 (4.7)	2 (1.3)	1 (0.7)		10 (6.7)	
* Economic status *							
High		1 (0.7)	0	0		1 (0.7)	0.866
Low		55 (36.7)	7 (4.7)	1 (0.7)		63 (42.0)	
Average		70 (46.6)	13 (8.6)	3 (2.0)		86 (57.3)	
*Overall HPLP II*							
Correct		83 (55.3)	17 (11.3)	2 (1.3)		102 (68.0)	0.173
Incorrect		43 (28.7)	3 (2.0)	2 (1.3)		48 (32.0)	
*Physical activity*							
Correct		78 (52.0)	17 (11.3)	3 (2.0)		98 (65.3)	0.120
Incorrect		48 (32.0)	3 (2.0)	1 (0.7)		52 (34.7)	
* Nutrition *							
Correct		99 (66.0)	18 (12.0)	2 (1.3)		119 (79.3)	0.171
Incorrect		27 (18.1)	2 (1.3)	2 (1.3)		31 (20.7)	
* Stress management *							
Correct		86 (57.3)	20 (13.4)	2 (1.3)		108 (72.0)	**0.008**
Incorrect		40 (26.7)	0	2 (1.3)		42 (28.0)	
**Predictors**		**Regression analysis, OR (95% CI)**					
	**CVD risk**	**High**	**Moderate**		**Low**		
* Age * ( * years * )							
Until 60		0.17 (0.05–0.57)	0.09 (0.02–0.43)		1.67 (0.15–19.1)		
61–70		0.72 (0.19–2.72)	0.42 (0.08–2.29)		2.29 (0.20–25.98)		
>70		Reference	Reference		Reference		
* Gender *							
Female		0.10 (0.04–0.29)	0.14 (0.05–0.42)		1.80 (1.73–1.84)		
Male		Reference	Reference		Reference		
* Stress management *							
Correct		0.20 (0.04–0.87)	0.0009 (0.0008–0.001)		2.65 (0.36–19.45)		
Incorrect		Reference	Reference		Reference		

**CVD**: Cardiovascular disease, **OR**: odds ratio, **CI**: confidence interval. The bold number are to emphasize the *p* value as significant.

## Data Availability

The data presented in this study are available on request from the corresponding author. The data are not publicly available for privacy reasons.

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
