# Peer review of "Association of Cardiovascular Disease Risk and Health-Related Behaviors in Stroke Patients"

_ijerph, 2023, doi:10.3390/ijerph20043693_

Round 1
Reviewer 1 Report
The aim of the manuscript is to assess stroke patients and the association between unhealthy behaviors (smoking, alcohol, unbalanced diet, sedentary, and stress) and an increased risk for cardiovascular disease. Lalo et al., conducted a cross-sectional study in the Vlora district regional hospital in Albania to accomplish this goal. They used specific criteria to enroll 150 patients. They report that 32% of patients manifest unhealthy behaviors and 84% of them have a high risk of cardiovascular disease. For men over 70 years old, this risk is higher. They conclude that in assessing stroke patients' training and education needs, healthcare professionals should also consider cardiovascular disease risk factors.
The manuscript is well written and the statistical analysis is carefully done. However, I have some minor issues that have to be addressed.
Abstract Line 13-15
For the most part, the abstract is a reasonable summary of the manuscript. However, within the abstract, it is stated that " The purpose is to identify strategies for prevention and improved management by patients and healthcare workers by assessing if stroke patients are more likely to develop cardiovascular disease if they engage in specific health-related behaviors”.
This statement is confusing, and it does not reflect the real aim of the manuscript. I suggest writing this part more clearly to highlight the real purpose. The aim of the work is expressed well in the final part of the introduction (Line 79-82).
Results Line 149-151 The journal instructions have been erroneously included in the manuscript. Please remove them.
Results Line 153-154 Are there some errors in the % of males reported? In the table is reported that 36% are females. Then I assume that the remaining 64% are men. If this is the case, please correct the number in the text.
Table 1 Is the mean age reported for both men and women? Can the authors report in the table the mean age of males and females separately?
Table 3 the values for F and F critical for age are the same. In the text is reported a significance that does not seem from these values. Is there an error? Please check table 3 information.
Results Line 174 Please change “Tables 4” in Table 4
Author Response
Responses to the reviewer's comments
We would like to thank Reviewers for taking the time and effort necessary to review the manuscript. We sincerely appreciate all valuable comments and suggestions, which helped us to improve the quality of the manuscript.
Appropriated changes, suggested by the Reviewer, has been introduced to the manuscript (highlighted in yellow within the document).
Comments to authors:
- This statement is confusing, and it does not reflect the real aim of the manuscript. I suggest writing this part more clearly to highlight the real purpose researchers…
Response 1:
We would like to thank the Reviewer for the comment. We reformulated the purpose of the study in the abstract section. Line 14–15 of revised manuscript.
Comments to authors:
- Results Line 149-151. The journal instructions have been erroneously included in the manuscript. Please remove them.
Response 2:
Thank you. We removed them. Line 149 of revised manuscript
Comments to authors:
- Results Line 153-154 Are there some errors in the % of males reported?.
Response 3:
Yes, you are right. I corrected it. Line 152 of revised manuscript
Comments to authors:
- Table 1. Can the authors report in the table the mean age of males and females separately?
Response 4:
Yes it`s done. See the table 1, line 156. .
Comments to authors:
- Table 3 the values for F and F critical for age are the same. In the text is reported a significance that does not seem from these values. Is there an error? Please check table 3 information
Response 5:
Yes there is an error. Thank you! I corrected it. Line 168 of revised manuscript.
Comments to authors:
- Results Line 174 Please change “Tables 4” in Table 4
Response 6:
Yes it`s done. Line 172 of revised manuscript.
Reviewer 2 Report
This manuscript presents an interesting set of findings. However, to approve it for the publication is necessary that the authors consider the following comments and to rewrite the results
Comments:
- Line 97: the authors must be indicate the choised value of significance level
- Line 164 reports the following text: “Overall, the HPLP II score was statistically significantly associated with all sociodemographic variables (Table 3)”. Where a significant statistical difference was observed, the analysis was continued by regression analysis”.
1) I suggest the authors to check the output of ANOVA, because, the overall HPLP II was not statistically significantly associated "with all sociodemographic variables" (Line 165) , but only the variable gender is statistically significantly associated with overall HPLP II (P=0.02). In line 141. in fact, the authors report as statistically significant the value p<0.05 ”
- Lines 174-178 report the following text: “The results presented in Table 4 found a statistically significant relationship between FRS score and variables such as age, gender, and stress management. It is noted that majority of patients belonging to the age group of 61–70 years old and 65% of women are more likely to be exposed to the risk of cardiovascular disease.”.
1)The authors can be better explain the logistic regression model(models)?
2) How have identified the confounders?
3) How is considered the CFI index in logistic regression model?
Author Response
We would like to thank the Reviewer for taking the time and effort necessary to review the manuscript. We sincerely appreciate all valuable comments and suggestions, which helped us to improve the quality of the manuscript.
Appropriated changes, suggested by the Reviewer, has been introduced to the manuscript (highlighted in yellow within the document).
Comments to authors:
- Line 97: the authors must be indicate the choised value of significance level
Response 1:
We added it. Line 97 of revised manuscript.
Comments to authors:
- Line 164 reports the following text: “Overall, the HPLP II score was statistically significantly associated with all sociodemographic variables (Table 3)”. I suggest the authors to check the output of ANOVA, because, the overall HPLP II was not statistically significantly associated "with all sociodemographic variables"……
Response 2:
We would like to thank the Reviewer for the suggestion. We checked the output of ANOVA to address the suggestion of the reviewer. There was an error for p value for the variable age. It was 0.05 and we corrected it as p=0.016. Where a significant statistical difference was observed, the analysis was continued by regression analysis. We removed the regression analysis for other non-significant relationships (the value p<0.05, F<F critical). The changed results are reflected in the results, discussion and conclusions section. Line 163, 168, 217–218, 279.
Comments to authors:
- Lines 174-178 report the following text: “The results presented in Table 4 found a statistically significant relationship between FRS score and variables such as age, gender, and stress management. It is noted that majority of patients belonging to the age group of 61–70 years old and 65% of women are more likely to be exposed to the risk of cardiovascular disease.”.
1)The authors can be better explain the logistic regression model(models)?
2) How have identified the confounders?
3) How is considered the CFI index in logistic regression model?
Response 3:
1) Univariate logistic regression was used line 144–145 of revised manuscript.
2) The confounders factors were not included in the statistical analysis since they weren`t part of the study objectives. This is the reason why are not mentioned into the manuscript.
3) Since there was no statistically significant relationship, we decided not to include it in the table. You can see the following table.
|
CVD Risk
|
Comorbidities |
Total |
p
|
||
|
≥3 |
<3 |
||||
|
The level |
High |
60 |
66 |
126 |
|
|
Moderate |
6 |
14 |
20 |
0.333 |
|
|
Low |
2 |
2 |
4 |
|
|
|
Total |
68 |
82 |
150 |
|
|
Reviewer 3 Report
Rezarta Lalo et al describes the association between Cardiovascular Disease risk and health related behaviors in Cerebrovascular Accident or 'Brain stroke' patients. The design and presentation of the manuscript is praiseworthy. Here are my concerns:
1. Line 62,63 say "in adult Albanian population, including that of the local population". Does "local population" include more than Albanians or "Adult" non-Albanians are excluded for such studies needs to clarify.
2. As the authors started with "brain or stroke" in line 33, it is better to keep the definition throughout the manuscript. As many of the readers might find it difficult to differentiate if the text is indicating brain stroke (CVA) or heart stroke (CVD). example line 90.
3. Line 94, "severe comorbidities" need examples.
4. Line 99, mentioned about "willingness" of the patients but it is important to include here if all the patients gave written consent for the study and publications as written in line 287.
5. Line 146 includes TG and FG without description of the abbreviation which has described later in line 160.
6. Line 149 to 151 are not clear. Are these instructions for the readers? If so, it needs to be rewritten with assertive tones. Examples, words "should" and "can be" need to be replaced.
7. Table 2,3 and 4 are excellent. If table 2 and 3 could manage in a single page that might make the manuscript much easier to navigate.
8. However Table 1 needs thoughts. Specially "less than high school" or "live alone" titles are confusing. It could be updated similar to table 3 where such titles got there own headings like "education" and "living" which made more sense.
9. The discussion could include more description of the association between Cardiovascular Disease risk and health related behavior without Brain Stroke. This inclusion will then validate the intension and importance of the association in Brain stroke patients.
Author Response
We would like to thank Reviewers for taking the time and effort necessary to review the manuscript. We sincerely appreciate all valuable comments and suggestions, which helped us to improve the quality of the manuscript.
Appropriated changes, suggested by the Reviewer, has been introduced to the manuscript (highlighted in yellow within the document).
Comments to authors:
- Line 62,63 say "in adult Albanian population, including that of the local population". Does "local population" include more than Albanians or "Adult" non-Albanians are excluded for such studies needs to clarify
Response 1:
Here, local population means the population of Vlora city, where we conducted the present study. However, in accordance with the reviewer's suggestion, we remove it.
Comments to authors:
- As the authors started with "brain or stroke" in line 33, it is better to keep the definition throughout the manuscript.
Response 2:
The text is indicating brain stroke. We keep the definition throughout the manuscript to address the suggestion of the reviewer. Line 12, 21, 38, 89, 161, 216, 277 of revised manuscript.
Comments to authors:
- Line 94, "severe comorbidities" need examples.
Response 3:
We would like to thank the Reviewer for the comment. Yes it`s done. We added the examples. Line 93, 94 of revised manuscript.
Comments to authors:
- Line 99, mentioned about "willingness" of the patients but it is important to include here if all the patients gave written consent for the study and publications as written in line 287.
Response 4:
Only the participants gave informed consent for the study and publications. We added the consent statement to address the suggestion of the reviewer in line 101 of revised manuscript.
Comments to authors:
- Line 146 includes TG and FG without description of the abbreviation which has described later in line 160.
Response 5:
Ok. We would like to thank the reviewer for the suggestion. We added description of the abbreviation. Line 147 of revised manuscript.
Comments to authors:
- Line 149 to 151 are not clear. Are these instructions for the readers? If so, it needs to be rewritten with assertive tones. Examples, words "should" and "can be" need to be replaced.
Response 6:
Please, could you clarify this comment once more. The comment is not related to the line number.
Comments to authors:
- Table 2,3 and 4 are excellent. If table 2 and 3 could manage in a single page that might make the manuscript much easier to navigate.
Response 7:
- Although I tried to manage both tables on the same page, it was impossible for the last row of table 3 due to its content.
Comments to authors:
- However Table 1 needs thoughts. Specially "less than high school" or "live alone" titles are confusing. It could be updated similar to table 3 where such titles got there own headings like "education" and "living" which made more sense.
Response 8:
- The reviewer's suggestion is addressed in table 1. Line 156 of revised manuscript.
Comments to authors:
- The discussion could include more description of the association between Cardiovascular Disease risk and health related behavior without Brain Stroke. This inclusion will then validate the intension and importance of the association in Brain stroke patients.
Response 9:
The reviewer's suggestion is addressed in discussion section. Line 234–243 of revised manuscript. We have added 2 references for more description of the association between Cardiovascular Disease risk and health related behavior without Brain Stroke. Line 377–381.